# Learning to Remember More with Less Memorization

**Hung Le, Truyen Tran and Svetha Venkatesh**
Applied AI Institute, Deakin University, Geelong, Australia
`{lethai,truyen.tran,svetha.venkatesh}@deakin.edu.au`

## Abstract

Memory-augmented neural networks consisting of a neural controller and an external memory have shown potentials in long-term sequential learning. Current RAM-like memory models maintain memory accessing every timesteps, thus they do not effectively leverage the short-term memory held in the controller. We hypothesize that this scheme of writing is suboptimal in memory utilization and introduces redundant computation. To validate our hypothesis, we derive a theoretical bound on the amount of information stored in a RAM-like system and formulate an optimization problem that maximizes the bound. The proposed solution dubbed Uniform Writing is proved to be optimal under the assumption of equal timestep contributions. To relax this assumption, we introduce modifications to the original solution, resulting in a solution termed Cached Uniform Writing. This method aims to balance between maximizing memorization and forgetting via overwriting mechanisms. Through an extensive set of experiments, we empirically demonstrate the advantages of our solutions over other recurrent architectures, claiming the state-of-the-arts in various sequential modeling tasks.

## 1 Introduction

A core task in sequence learning is to capture long-term dependencies amongst timesteps which demands memorization of distant inputs. In recurrent neural networks (RNNs), the memorization is implicitly executed via integrating the input history into the state of the networks. However, learning vanilla RNNs over long distance proves to be difficult due to the vanishing gradient problem (Bengio et al., 1994; Pascanu et al., 2013). One alleviation is to introduce skip-connections along the execution path, in the forms of dilated layers (Van Den Oord et al., 2016; Chang et al., 2017), attention mechanisms (Bahdanau et al., 2015; Vaswani et al., 2017) and external memory (Graves et al., 2014; 2016).

Amongst all, using external memory most resembles human cognitive architecture where we perceive the world sequentially and make decision by consulting our memory. Recent attempts have simulated this process by using RAM-like memory architectures that store information into memory slots. Reading and writing are governed by neural controllers using attention mechanisms. These memory-augmented neural networks (MANN) have demonstrated superior performance over recurrent networks in various synthetic experiments (Graves et al., 2016) and realistic applications (Le et al., 2018a;b; Franke et al., 2018).

Despite the promising empirical results, there is no theoretical analysis or clear understanding on optimal operations that a memory should have to maximize its performance. To the best of our knowledge, no solution has been proposed to help MANNs handle ultra-long sequences given limited memory. This scenario is practical because (i) sequences in the real-world can be very long while the computer resources are limited and (ii) it reflects the ability to compress in human brain to perform life-long learning. Previous attempts such as (Rae et al., 2016) try to learn ultra-long sequences by expanding the memory, which is not always feasible and do not aim to optimize the memory by some theoretical criterion. This

paper presents a new approach towards finding optimal operations for MANNs that serve the purpose of learning longer sequences with finite memory.

More specifically, upon analyzing RNN and MANN operations we first introduce a measurement on the amount of information that a MANN holds after encoding a sequence. This metric reflects the quality of memorization under the assumption that contributions from timesteps are equally important. We then derive a generic solution to optimize the measurement. We term this optimal solution as Uniform Writing (UW), and it is applicable for any MANN due to its generality. Crucially, UW helps reduce significantly the computation time of MANN. Third, to relax the assumption and enable the method to work in realistic settings, we further propose Cached Uniform Writing (CUW) as an improvement over the Uniform Writing scheme. By combining uniform writing with local attention, CUW can learn to discriminate timesteps while maximizing local memorization. Finally we demonstrate that our proposed models outperform several MANNs and other state-of-the-art methods in various synthetic and practical sequence modeling tasks.

## 2 Methods

### 2.1 Theoretical Analysis

Memory-augmented neural networks can be viewed as an extension of RNNs with external memory $M$. The memory supports read and write operations based on the output $o_t$ of the controller, which in turn is a function of current timestep input $x_t$, previous hidden state $h_{t-1}$ and read value $r_{t-1}$ from the memory. Let assume we are given these operators from recent MANNs such as NTM (Graves et al., 2014) or DNC (Graves et al., 2016), represented as:

$$r_t = f_r\left(o_t, M_{t-1}\right) \qquad (1) \qquad\qquad M_t = f_w\left(o_t, M_{t-1}\right) \qquad (2)$$

The controller output and hidden state are updated as follows:

$$o_t = f_o\left(h_{t-1}, r_{t-1}, x_t\right) \qquad (3) \qquad\qquad h_t = f_h\left(h_{t-1}, r_{t-1}, x_t\right) \qquad (4)$$

Here, $f_o$ and $f_h$ are often implemented as RNNs while $f_r$ and $f_w$ are designed specifically for different memory types.

Current MANNs only support regular writing by applying Eq. (2) every timestep. In effect, regular writing ignores the accumulated short-term memory stored in the controller hidden states which may well-capture the recent subsequence. We argue that the controller does not need to write to memory continuously as its hidden state also supports memorizing. Another problem of regular writing is time complexity. As the memory access is very expensive, reading/writing at every timestep makes MANNs much slower than RNNs. This motivates a irregular writing strategy to utilize the memorization capacity of the controller and consequently, speed up the model. In the next sections, we first define a metric to measure the memorization performance of RNNs, as well as MANNs. Then, we solve the problem of finding the best irregular writing that optimizes the metric.

### 2.1.1 Memory analysis of RNNs

We first define the ability to "remember" of recurrent neural networks, which is closely related to the vanishing/exploding gradient problem (Pascanu et al., 2013). In RNNs, the state transition $h_t = \phi\left(h_{t-1}, x_t\right)$ contains contributions from not only $x_t$, but also previous timesteps $x_{i<t}$ embedded in $h_{t-1}$. Thus, $h_t$ can be considered as a function of timestep inputs, i.e, $h_t = f\left(x_1, x_2, ..., x_t\right)$. One way to measure how much an input $x_i$ contributes to the value of $h_t$ is to calculate the norm of the gradient $\left\|\frac{\partial h_t}{\partial x_i}\right\|$. If the norm equals zero, $h_t$ is constant w.r.t $x_i$, that is, $h_t$ does not "remember" $x_i$. As a bigger $\left\|\frac{\partial h_t}{\partial x_i}\right\|$ implies more influence of $x_i$ on $h_t$, we propose using $\left\|\frac{\partial h_t}{\partial x_i}\right\|$ to measure the contribution of the $i$-th input

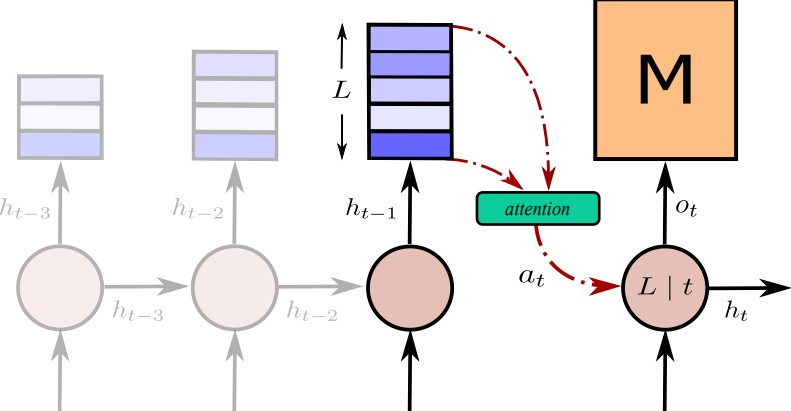

Figure 1: Writing mechanism in Cached Uniform Writing. During non-writing intervals, the controller hidden states are pushed into the cache. When the writing time comes, the controller attends to the cache, chooses suitable states and accesses the memory. The cache is then emptied.

to the $t$-th hidden state. Let $c_{i,t}$ denotes this term, we can show that in the case of common RNNs, $\lambda_c c_{i,t} \geq c_{i-1,t}$ with some $\lambda_c \in \mathbb{R}^+$ (see Appendix A - C for proof). This means further to the past, the contribution decays (when $\lambda_c < 1$) or grows (when $\lambda_c > 1$) with the rate of at least $\lambda_c$. We can measure the average amount of contributions across $T$ timesteps as follows (see Appendix D for proof):

**Theorem 1.** *There exists $\lambda \in \mathbb{R}^+$ such that the average contribution of a sequence of length $T$ with respect to a RNN can be quantified as the following:*

$$
I_\lambda = \frac{\sum_{t=1}^{T} c_{t,T}}{T} = c_{T,T} \frac{\sum_{t=1}^{T} \lambda^{T-t}}{T}
\tag{5}
$$

If $\lambda < 1$, $\lambda^{T-t} \to 0$ as $T - t \to \infty$. This is closely related to vanishing gradient problem. LSTM is known to "remember" long sequences better than RNN by using extra memory gating mechanisms, which help $\lambda$ to get closer to 1. If $\lambda > 1$, the system may be unstable and suffer from the exploding gradient problem.

### 2.1.2 MEMORY ANALYSIS OF MANNS

In slot-based MANNs, memory $M$ is a set of $D$ memory slots. A write at step $t$ can be represented by the controller's hidden state $h_t$, which accumulates inputs over several timesteps (i.e., $x_1, ...,x_t$). If another write happens at step $t+k$, the state $h_{t+k}$'s information containing timesteps $x_{t+1}, ...,x_{t+k}$ is stored in the memory ($h_{t+k}$ may involves timesteps further to the past, yet they are already stored in the previous write and can be ignored). During writing, overwriting may happen, replacing an old write with a new one. Thus after all, $D$ memory slots associate with $D$ chosen writes of the controller. From these observations, we can generalize Theorem 1 to the case of MANNs having $D$ memory slots (see Appendix E for proof).

**Theorem 2.** *With any $D$ chosen writes at timesteps $1 \leq K_1 < K_2 < ... < K_D < T$, there exist $\lambda, C \in \mathbb{R}^+$ such that the lower bound on the average contribution of a sequence of length $T$ with respect to a MANN having $D$ memory slots can be quantified as the following:*

---

**Algorithm 1** Cached Uniform Writing

---
**Require:** a sequence $x = \{x_t\}_{t=1}^T$, a cache $C$ sized $L$, a memory sized $D$.
 1: **for** $t = 1, T$ **do**
 2:     $C.\text{append}(h_{t-1})$
 3:     **if** $t \bmod L == 0$ **then**
 4:         Use Eq.(10) to calculate $a_t$
 5:         Execute Eq.(3): $o_t = f_o\left(a_t, r_{t-1}, x_t\right)$
 6:         Execute Eq.(4): $h_t = f_h\left(a_t, r_{t-1}, x_t\right)$
 7:         Update the memory using Eq.(2)
 8:         Read $r_t$ from the memory using Eq.(1)
 9:         $C.\text{clear}()$
10:     **else**
11:         Update the controller using Eq.(4): $h_t = f_h\left(h_{t-1}, r_{t-1}, x_t\right)$
12:         Assign $r_t = r_{t-1}$
13:     **end if**
14: **end for**

---

$$I_\lambda = C \frac{\sum_{t=1}^{K_1} \lambda^{K_1-t} + \sum_{t=K_1+1}^{K_2} \lambda^{K_2-t} + ... + \sum_{t=K_{D-1}+1}^{K_D} \lambda^{K_D-t} + \sum_{t=K_D+1}^{T} \lambda^{T-t}}{T}$$

$$= \frac{C}{T} \sum_{i=1}^{D+1} \sum_{j=0}^{l_i-1} \lambda^j = \frac{C}{T} \sum_{i=1}^{D+1} f_\lambda(l_i) \tag{6}$$

$$\textit{where } l_i = \begin{cases} K_1 & ; i = 1 \\ K_i - K_{i-1} & ; D \geq i > 1 \\ T - K_D & ; i = D+1 \end{cases}, \; f_\lambda(x) = \begin{cases} \frac{1-\lambda^x}{1-\lambda} & \lambda \neq 1 \\ x & \lambda = 1 \end{cases}, \; \forall x \in \mathbb{R}^+.$$

If $\lambda \leq 1$, we want to maximize $I_\lambda$ to keep the information from vanishing. On the contrary, if $\lambda > 1$, we may want to minimize $I_\lambda$ to prevent the information explosion. As both scenarios share the same solution (see Appendix F), thereafter we assume that $\lambda \leq 1$ holds for other analyses. By taking average over $T$, we are making an assumption that all timesteps are equally important. This helps simplify the measurement as $I_\lambda$ is independent of the specific position of writing. Rather, it is a function of the interval lengths between the writes. This turns out to be an optimization problem whose solution is stated in the following theorem.

**Theorem 3.** *Given $D$ memory slots, a sequence with length $T$, a decay rate $0 < \lambda \leq 1$, then the optimal intervals $\{l_i \in \mathbb{R}^+\}_{i=1}^{D+1}$ satisfying $T = \sum_{i=1}^{D+1} l_i$ such that the lower bound on the average contribution $I_\lambda = \frac{C}{T} \sum_{i=1}^{D+1} f_\lambda(l_i)$ is maximized are the following:*

$$l_1 = l_2 = ... = l_{D+1} = \frac{T}{D+1} \tag{7}$$

We name the optimal solution as Uniform Writing (UW) and refer to the term $\frac{T}{D+1}$ and $\frac{D+1}{T}$ as the *optimal interval* and the *compression ratio*, respectively. The proof is given in Appendix F.

## 2.2 PROPOSED MODELS

Uniform writing can apply to any MANNs that support writing operations. Since the writing intervals are discrete, i.e., $l_i \in \mathbb{N}^+$, UW is implemented as the following:

$$M_t = \begin{cases} f_w\left(o_t, M_{t-1}\right) & if\ t = \left\lfloor \frac{T}{D+1} \right\rfloor k, k \in \mathbb{N}^+ \\ M_{t-1} & otherwise \end{cases} \tag{8}$$

By following Eq. (8), the write intervals are close to the optimal interval defined in Theorem 3 and approximately maximize the average contribution. This writing policy works well if timesteps are equally important and the task is to remember all of them to produce outputs (i.e., in copy task). However, in reality, timesteps are not created equal and a good model may need to ignore unimportant or noisy timesteps. That is why overwriting in MANN can be necessary. In the next section, we propose a method that tries to balance between following the optimal strategy and employing overwriting mechanism as in current MANNs.

### 2.2.1 Local optimal design

To relax the assumptions of Theorem 3, we propose two improvements of the Uniform Writing (UW) strategy. First, the intervals between writes are equal with length $L$ ($1 \leq L \leq \left\lfloor \frac{T}{D+1} \right\rfloor$). If $L = 1$, the strategy becomes regular writing and if $L = \left\lfloor \frac{T}{D+1} \right\rfloor$, it becomes uniform writing. This ensures that after $\left\lfloor \frac{T}{L} \right\rfloor$ writes, all memory slots should be filled and the model has to learn to overwrite. Meanwhile, the average kept information is still locally maximized every $L * D$ timesteps.

Second, we introduce a cache of size $L$ to store the hidden states of the controller during a write interval. Instead of using the hidden state at the writing timestep to update the memory, we perform an attention over the cache to choose the best representative hidden state. The model will learn to assign attention weights to the elements in the cache. This mechanism helps the model consider the importance of each timestep input in the local interval and thus relax the equal contribution assumption of Theorem 3. We name the writing strategy that uses the two mentioned-above improvements as Cached Uniform Writing (CUW). An illustration of the writing mechanism is depicted in Fig. 1.

### 2.2.2 Local memory-augmented attention unit

In this subsection, we provide details of the attention mechanism used in our CUW. To be specific, the best representative hidden state $a_t$ is computed as follows:

$$\alpha_{tj} = softmax\left(v^T \tanh\left(Wh_{t-1} + Ud_j + Vr_{t-1}\right)\right) \quad (9) \qquad\qquad a_t = \sum_{j=1}^{L} \alpha_{tj} d_j \qquad (10)$$

where $\alpha_{tj}$ is the attention score between the $t$-th writing step and the $j$-th element in the cache; $W$, $U$, $V$ and $v$ are parameters; $h$ and $r$ are the hidden state of the controller and the read-out (Eq. (1)), respectively; $d_j$ is the cache element and can be implemented as the controller's hidden state ($d_j = h_{t-1-L+j}$).

The vector $a_t$ will be used to replace the previous hidden state in updating the controller and memory. The whole process of performing CUW is summarized in Algo. 1.

## 3 Results

### 3.1 An Ablation Study: Memory-augmented Neural Networks with and without Uniform Writing

In this section, we study the impact of uniform writing on MANNs under various circumstances (different controller types, memory types and number of memory slots). We restrict the memorization problem to the double task in which the models must reconstruct a sequence of integers sampled uniformly from range $[1, 10]$ twice. We cast this problem to a sequence to sequence problem with 10 possible outputs per decoding step. The training

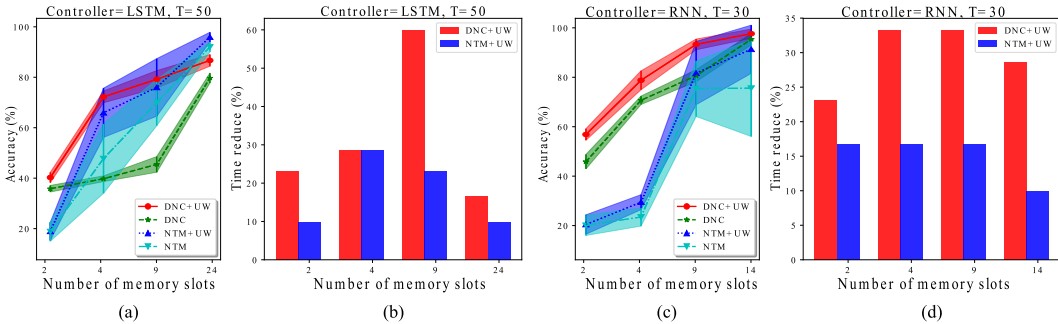

Figure 2: The accuracy (%) and computation time reduction (%) with different memory types and number of memory slots. The controllers/sequence lengths/memory sizes are chosen as LSTM/50/$\{2, 4, 9, 24\}$ (a&b) and RNN/30/$\{2, 4, 9, 14\}$ (c&d), respectively.

stops after 10,000 iterations of batch size 64. We choose DNC[1] and NTM[2] as the two MANNs in the experiment. The recurrent controllers can be RNN or LSTM. With LSTM controller, the sequence length is set to 50. We choose sequence length of 30 to make it easier for the RNN controller to learn the task. The number of memory slots $D$ is chosen from the set $\{2, 4, 9, 24\}$ and $\{2, 4, 9, 14\}$ for LSTM and RNN controllers, respectively. More memory slots will make UW equivalent to the regular writing scheme. For this experiment, we use Adam optimizer (Kingma & Ba, 2014) with initial learning rate and gradient clipping of $\{0.001, 0.0001\}$ and $\{1, 5, 10\}$, respectively. The metric used to measure the performance is the average accuracy across decoding steps. For each configuration of hyper-parameters, we run the experiment 5 times and report the mean accuracy with error bars.

Figs. 2(a) and (c) depict the performance of UW and regular writing under different configurations. In any case, UW boosts the prediction accuracy of MANNs. The performance gain can be seen clearly when the compression ratio is between $10 - 40\%$. This is expected since when the compression ratio is too small or too big, UW converges to regular writing. Interestingly, increasing the memory size does not always improve the performance, as in the case of NTM with RNN controllers. Perhaps, learning to attend to many memory slots is tricky for some task given limited amount of training data. This supports the need to apply UW to MANN with moderate memory size. We also conduct experiments to verify the benefit of using UW for bigger memory. The results can be found in Appendix H.

We also measure the speed-up of training time when applying UW on DNC and NTM, which is illustrated in Figs. 2(b) and (d). The result shows that with UW, the training time can drop up to 60% for DNC and 28% for NTM, respectively. As DNC is more complicated than NTM, using UW to reduce memory access demonstrates clearer speed-up in training (similar behavior can be found for testing time).

## 3.2 SYNTHETIC MEMORIZATION

Here we address a broader range of baselines on two synthetic memorization tasks, which are the sequence copy and reverse. In these tasks, there is no discrimination amongst timesteps so the model's goal is to learn to compress the input efficiently for later retrieval. We experiment with different sequence lengths of 50 and 100 timesteps. Other details are the same as the previous double task except that we fix the learning rate and gradient clipping to 0.001 and 10, respectively. The standard baselines include LSTM, NTM and DNC. All memory-augmented models have the same memory size of 4 slots, corresponding to compression ratio of 10% and 5%, respectively. We aim at this range of compression ratio to match harsh practical requirements. UW and CUW (cache size $L = 5$) are built upon the DNC, which from our previous observations, works best for given compression ratios. We choose different dimensions $N_h$ for the hidden vector of the controllers to ensure the model

---

[1]Our reimplementation based on https://github.com/deepmind/dnc

[2]https://github.com/MarkPKCollier/NeuralTuringMachine

| Model | $N_h$ | # parameter | Copy | | Reverse | |
|---|---|---|---|---|---|---|
| | | | L=50 | L=100 | L=50 | L=100 |
| LSTM | 125 | 103,840 | 15.6 | 12.7 | 49.6 | 26,1 |
| NTM | 100 | 99,112 | 40.1 | 11.8 | 61.1 | 20.3 |
| DNC | 100 | 98,840 | 68.0 | 44.2 | 65.0 | 54.1 |
| DNC+RW | 100 | 98,840 | 47.6 | 37.0 | 70.8 | 50.1 |
| DNC+UW | 100 | 98,840 | **97.7** | **69.3** | **100** | **79.5** |
| DNC+CUW | 95 | 96,120 | 83.8 | 55.7 | 93.3 | 55.4 |

Table 1: Test accuracy (%) on synthetic memorization tasks. MANNs have 4 memory slots.

sizes are approximately equivalent. To further verify that our UW is actually the optimal writing strategy, we design a new baseline, which is DNC with random irregular writing strategy (RW). The write is sampled from a binomial distribution with $p = (D+1)/T$ (equivalent to compression ratio). After sampling, we conduct the training for that policy. The final performances of RW are taken average from 3 different random policies' results.

The performance of the models is listed in Table 1. As clearly seen, UW is the best performer for the pure memorization tests. This is expected from the theory as all timesteps are importantly equivalent. Local attention mechanism in CUW does not help much in this scenario and thus CUW finishes the task as the runner-up. Reverse seems to be easier than copy as the models tend to "remember" more the last-seen timesteps whose contributions $\lambda^{T-t}$ remains significant. In both cases, other baselines including random irregular and regular writing underperform our proposed models by a huge margin.

## 3.3 SYNTHETIC REASONING

Tasks in the real world rarely involve just memorization. Rather, they require the ability to selectively remember the input data and synthesize intermediate computations. To investigate whether our proposed writing schemes help the memory-augmented models handle these challenges, we conduct synthetic reasoning experiments which include add and max tasks. In these tasks, each number in the output sequence is the sum or the maximum of two numbers in the input sequence. The pairing is fixed as: $y_t = \frac{x_t + x_{T-t}}{2}, t = \overline{1, \lfloor \frac{T}{2} \rfloor}$ for add task and $y_t = \max(x_{2t}, x_{2t+1}), t = \overline{1, \lfloor \frac{T}{2} \rfloor}$ for max task, respectively. The length of the output sequence is thus half of the input sequence. A brief overview of input/output format for these tasks can be found in Appendix G. We deliberately use local (max) and distant (add) pairing rules to test the model under different reasoning strategies. The same experimental setting as in the previous section is applied except for the data sample range for the max task, which is $[1, 50]$[3]. LSTM and NTM are excluded from the baselines as they fail on these tasks.

Table 2 shows the testing results for the reasoning tasks. Since the memory size is small compared to the number of events, regular writing or random irregular writing cannot compete with the uniform-based writing policies. Amongst all baselines, CUW demonstrates superior performance in both tasks thanks to its local attention mechanism. It should be noted that the timesteps should not be treated equally in these reasoning tasks. The model should weight a timestep differently based on either its content (max task) or location (add task) and maintain its memory for a long time by following uniform criteria. CUW is designed to balance the two approaches and thus it achieves better performance. Further insights into memory operations of these models are given in Appendix I.

## 3.4 SYNTHETIC SINUSOIDAL REGRESSION

In real-world settings, sometimes a long sequence can be captured and fully reconstructed by memorizing some of its feature points. For examples, a periodic function such as sinusoid can be well-captured if we remember the peaks of the signal. By observing the peaks,

---

[3]With small range like $[1, 10]$, there is no much difference in performance amongst models

| Model | Add | | Max | |
|---|---|---|---|---|
| | L=50 | L=100 | L=50 | L=100 |
| DNC | 83.8 | 22.3 | 59.5 | 27.4 |
| DNC+RW | 83.0 | 22.7 | 59.7 | 36.5 |
| DNC+UW | 84.8 | 50.9 | 71.7 | 66.2 |
| DNC+CUW | **94.4** | **60.1** | **82.3** | **70.7** |

Table 2: Test accuracy (%) on synthetic reasoning tasks. MANNs have 4 memory slots.

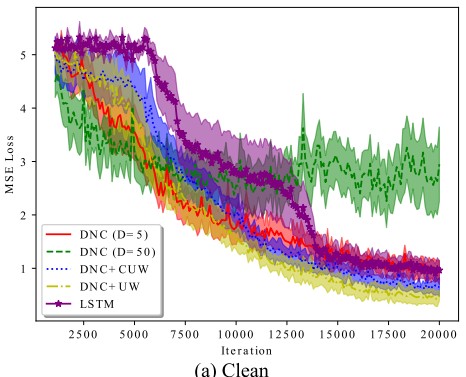
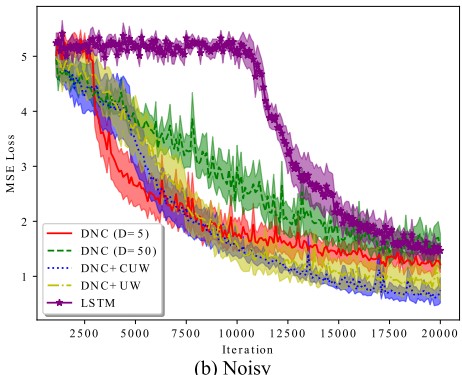

(a) Clean           (b) Noisy

Figure 3: Learning curves of models in clean (a) and noisy (b) sinusoid regression experiment.

we can deduce the frequency, amplitude, phase and thus fully reconstructing the function. To demonstrate that UW and CUW are useful for such scenarios, we design a sequential continuation task, in which the input is a sequence of sampling points across some sinusoid: $y = 5 + A\sin(2\pi fx + \varphi)$. Here, $A \sim \mathcal{U}(1,5)$, $f \sim \mathcal{U}(10,30)$ and $\varphi \sim \mathcal{U}(0,100)$. After reading the input $y = \{y_t\}_{t=1}^{T}$, the model have to generate a sequence of the following points in the sinusoid. To ensure the sequence $y$ varies and covers at least one period of the sinusoid, we set $x = \{x_t\}_{t=1}^{T}$ where $x_i = (t + \epsilon_1)/1000$, $\epsilon_1 \sim \mathcal{U}(-1,1)$. The sequence length for both input and output is fixed to $T = 100$. The experimental models are LSTM, DNC, UW and CUW (built upon DNC). For each model, optimal hyperparameters including learning rate and clipping size are tuned with 10,000 generated sinusoids. The memories have 4 slots and all baselines have similar parameter size. We also conduct the experiment with noisy inputs by adding a noise $\epsilon_2 \sim \mathcal{U}(-2,2)$ to the input sequence $y$. This increases the difficulty of the task. The loss is the average of mean square error (MSE) over decoding timesteps.

We plot the mean learning curves with error bars over 5 runnings for sinusoidal regression task under clean and noisy condition in Figs. 3(a) and (b), respectively. Regular writing DNC learns fast at the beginning, yet soon saturates and approaches the performance of LSTM ($MSE = 1.05$ and $1.39$ in clean and noisy condition, respectively). DNC performance does not improve much as we increase the memory size to 50, which implies the difficulty in learning with big memory. Although UW starts slower, it ends up with lower errors than DNC and perform slightly better than CUW in clean condition ($MSE = 0.44$ for UW and $0.61$ for CUW). CUW demonstrates competitive performance against other baselines, approaching to better solution than UW for noisy task where the model should discriminate the timesteps ($MSE = 0.98$ for UW and $0.55$ for CUW). More visualizations can be found in Appendix J.

### 3.5 FLATTEN IMAGE RECOGNITION

We want to compare our proposed models with DNC and other methods designed to help recurrent networks learn longer sequence. The chosen benchmark is a pixel-by-pixel image classification task on MNIST in which pixels of each image are fed into a recurrent model sequentially before a prediction is made. In this task, the sequence length is fixed to 768

| Model | MNIST | pMNIST |
|---|---|---|
| iRNN[†] | 97.0 | 82.0 |
| uRNN° | 95.1 | 91.4 |
| r-LSTM Full BP* | 98.4 | 95.2 |
| Dilated-RNN♦ | 95.5 | 96.1 |
| Dilated-GRU♦ | **99.2** | 94.6 |
| DNC | 98.1 | 94.0 |
| DNC+UW | 98.6 | 95.6 |
| DNC+CUW | 99.1 | **96.3** |

Table 3: Test accuracy (%) on MNIST, pMNIST. Previously reported results are from (Le et al., 2015)[†], (Arjovsky et al., 2016)°, (Trinh et al., 2018)*, and (Chang et al., 2017)♦.

with highly redundant timesteps (black pixels). The training, validation and testing sizes are 50,000, 10,000 and 10,000, respectively. We test our models on both versions of non-permutation (MNIST) and permutation (pMNIST). More details on the task and data can be found in (Le et al., 2015). For DNC, we try with several memory slots from $\{15, 30, 60\}$ and report the best results. For UW and CUW, memory size is fixed to 15 and cache size $L$ is set to 10. The controllers are implemented as single layer GRU with 100-dimensional hidden vector. To optimize the models, we use RMSprop with initial learning rate of 0.0001.

Table 3 shows that DNC underperforms r-LSTM, which indicates that regular DNC with big memory finds it hard to beat LSTM-based methods. After applying UW, the results get better and with CUW, it shows significant improvement over r-LSTM and demonstrates competitive performance against dilated-RNNs models. Notably, dilated-RNNs use 9 layers in their experiments compared to our singer layer controller. Furthermore, our models exhibit more consistent performance than dilated-RNNs. For completeness, we include comparisons between CUW and non-recurrent methods in Appendix K

### 3.6 DOCUMENT CLASSIFICATION

To verify our proposed models in real-world applications, we conduct experiments on document classification task. In the task, the input is a sequence of words and the output is the classification label. Following common practices in (Yogatama et al., 2017; Seo et al., 2018), each word in the document is embedded into a 300-dimensional vector using Glove embedding (Pennington et al., 2014). We use RMSprop for optimization, with initial learning rate of 0.0001. Early-stop training is applied if there is no improvement after 5 epochs in the validation set. Our UW and CUW are built upon DNC with single layer 512-dimensional LSTM controller and the memory size is chosen in accordance with the average length of the document, which ensures $10 - 20\%$ compression ratio. The cache size for CUW is fixed to 10. The datasets used in this experiment are common big datasets where the number of documents is between 120,000 and 1,400,000 with maximum of 4,392 words per document (see Appendix L for further details). The baselines are recent state-of-the-arts in the domain, some of which are based on recurrent networks such as D-LSTM (Yogatama et al., 2017) and Skim-LSTM (Seo et al., 2018). We exclude DNC from the baselines as it is inefficient to train the model with big document datasets.

Our results are reported in Table 4. On five datasets out of six, our models beat or match the best published results. For IMDb dataset, our methods outperform the best recurrent model (Skim-LSTM). The performance gain is competitive against that of the state-of-the-arts. In most cases, CUW is better than UW, which emphasizes the importance of relaxing the timestep equality assumption in practical situations. Details results across different runs for our methods are listed in Appendix M.

---

[4]Methods that use semi-supervised training to achieve higher accuracy are not listed.

| Model | AG | IMDb[4] | Yelp P. | Yelp F. | DBP | Yah. A. |
|---|---|---|---|---|---|---|
| VDCNN[•] | 91.3 | - | 95.7 | 64.7 | 98.7 | 73.4 |
| D-LSTM[*] | - | - | 92.6 | 59.6 | 98.7 | *73.7* |
| Standard LSTM[‡] | 93.5 | 91.1 | - | - | - | - |
| Skim-LSTM[‡] | *93.6* | 91.2 | - | - | - | - |
| Region Embedding[▲] | 92.8 | - | ***96.4*** | *64.9* | *98.9* | *73.7* |
| DNC+UW | 93.7 | **91.4** | **96.4** | 65.3 | **99.0** | 74.2 |
| DNC+CUW | **93.9** | 91.3 | **96.4** | **65.6** | **99.0** | **74.3** |

Table 4: Document classification accuracy (%) on several datasets. Previously reported results are from (Conneau et al., 2016)[•], (Yogatama et al., 2017)[*], (Seo et al., 2018)[‡] and (Qui et al., 2018)[▲]. We use italics to denote the best published and bold the best records.

## 4 Related Work

Traditional recurrent models such as RNN/LSTM (Elman, 1990; Hochreiter & Schmidhuber, 1997) exhibit some weakness that prevent them from learning really long sequences. The reason is mainly due to the vanishing gradient problem (Pascanu et al., 2013) or to be more specific, the exponential decay of input value over time. One way to overcome this problem is enforcing the exponential decay factor close to one by putting a unitary constraint on the recurrent weight (Arjovsky et al., 2016; Wisdom et al., 2016). Although this approach is theoretically motivated, it restricts the space of learnt parameters.

More relevant to our work, the idea of using less or adaptive computation for good has been proposed in (Graves, 2016; Yu et al., 2017; 2018; Seo et al., 2018). Most of these works are based on the assumption that some of timesteps in a sequence are unimportant and thus can be ignored to reduce the cost of computation and increase the performance of recurrent networks. Different form our approach, these methods lack theoretical supports and do not directly aim to solve the problem of memorizing long-term dependencies.

Dilated RNN (Chang et al., 2017) is another RNN-based proposal which improves long-term learning by stacking multiple dilated recurrent layers with hierarchical skip-connections. This theoretically guarantees the mean recurrent length and shares with our method the idea to construct a measurement on memorization capacity of the system and propose solutions to optimize it. The difference is that our system is memory-augmented neural networks while theirs is multi-layer RNNs, which leads to totally different optimization problems.

Recent researches recommend to replace traditional recurrent models by other neural architectures to overcome the vanishing gradient problem. The Transformer (Vaswani et al., 2017) attends to all timesteps at once, which ensures instant access to distant timestep yet requires quadratic computation and physical memory proportional to the sequence length. Memory-augmented neural networks (MANNs), on the other hand, learn to establish a limited-size memory and attend to the memory only, which is scalable to any-length sequence. Compared to others, MANNs resemble both computer architecture design and human working memory (Logie, 2014). However, the current understanding of the underlying mechanisms and theoretical foundations for MANN are still limited.

Recent works on MANN rely almost on reasonable intuitions. Some introduce new addressing mechanisms such as location-based (Graves et al., 2014), least-used (Santoro et al., 2016) and order-based (Graves et al., 2016). Others focus on the scalability of MANN by using sparse memory access to avoid attending to a large number of memory slots (Rae et al., 2016). These problems are different from ours which involves MANN memorization capacity optimization.

Our local optimal solution to this problem is related to some known neural caching (Grave et al., 2017b;a; Yogatama et al., 2018) in terms of storing recent hidden states for later encoding uses. These methods either aim to create structural bias to ease the learning process (Yogatama et al., 2018) or support large scale retrieval (Grave et al., 2017a). These are different from our caching purpose, which encourages overwriting and relaxes the equal

contribution assumption of the optimal solution. Also, the details of implementation are different as ours uses local memory-augmented attention mechanisms.

## 5  Conclusions

We have introduced Uniform Writing (UW) and Cached Uniform Writing (CUW) as faster solutions for longer-term memorization in MANNs. With a comprehensive suite of synthetic and practical experiments, we provide strong evidences that our simple writing mechanisms are crucial to MANNs to reduce computation complexity and achieve competitive performance in sequence modeling tasks. In complement to the experimental results, we have proposed a meaningful measurement on MANN memory capacity and provided theoretical analysis showing the optimality of our methods. Further investigations to tighten the measurement bound will be the focus of our future work.

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

## Appendix

### A  Derivation on the bound inequality in linear dynamic system

The linear dynamic system hidden state is described by the following recursive equation:

$$h_t = Wx_t + Uh_{t-1} + b$$

By induction,

$$h_t = \sum_{i=1}^{t} U^{t-i}Wx_i + C$$

where $C$ is some constant with respect to $x_i$. In this case, $\frac{\partial h_t}{\partial x_i} = U^{t-i}W$. By applying norm sub-multiplicativity[5],

$$
\begin{aligned}
c_{i-1,t} &= \left\| U^{t-i+1}W \right\| \\
&\leq \|U\| \left\| U^{t-i}W \right\| \\
&= \|U\| \, c_{i,t}
\end{aligned}
$$

That is, $\lambda_c = \|U\|$.

### B  Derivation on the bound inequality in standard RNN

The standard RNN hidden state is described by the following recursive equation:

$$h_t = \tanh(Wx_t + Uh_{t-1} + b)$$

From $\frac{\partial h_t}{\partial x_i} = \frac{\partial h_t}{\partial h_{t-1}} \frac{\partial h_{t-1}}{\partial x_i}$, by induction,

$$\frac{\partial h_t}{\partial x_i} = \left( \prod_{j=i+1}^{t} \frac{\partial h_j}{\partial h_{j-1}} \right) \frac{\partial h_i}{\partial x_i} = \left( \prod_{j=i}^{t} diag\left( \tanh'(a_j) \right) \right) U^{t-i}W$$

where $a_j = Wx_j + Uh_{j-1} + b$ and $diag\left( \cdot \right)$ converts a vector into a diagonal matrix. As $0 \leq \tanh'(x) = 1 - \tanh(x)^2 \leq 1$, $\left\| diag\left( \tanh'(X) \right) \right\|$ is bounded by some value $B$. By applying norm sub-multiplicativity,

$$
\begin{aligned}
c_{i-1,t} &= \left\| \left[ \prod_{j=i}^{t} diag\left( \tanh'(a_j) \right) \right] diag\left( \tanh'(a_{i-1}) \right) U^{t-i+1}W \right\| \\
&\leq \|U\| \left\| \prod_{j=i}^{t} diag\left( \tanh'(a_j) \right) U^{t-i}W \right\| \left\| diag\left( \tanh'(a_{i-1}) \right) \right\| \\
&= B \|U\| \, c_{i,t}
\end{aligned}
$$

That is, $\lambda_c = B \|U\|$.

---

[5]If not explicitly stated otherwise, norm refers to any consistent matrix norm which satisfies sub-multiplicativity.

## C  Derivation on the bound inequality in LSTM

For the case of LSTM, the recursive equation reads:

$$
\begin{aligned}
c_t &= \sigma\left(U_f x_t + W_f h_{t-1} + b_f\right) \odot c_{t-1} \\
&\quad + \sigma\left(U_i x_t + W_i h_{t-1} + b_i\right) \odot \tanh\left(U_z x_t + W_z h_{t-1} + b_z\right) \\
h_t &= \sigma\left(U_o x_t + W_o h_{t-1} + b_o\right) \odot \tanh\left(c_t\right)
\end{aligned}
$$

Taking derivatives,

$$
\begin{aligned}
\frac{\partial h_j}{\partial h_{j-1}} &= \sigma'\left(o_j\right)\tanh\left(c_j\right)W_o + \sigma\left(o_j\right)\tanh'\left(c_j\right)\sigma'\left(f_j\right)c_{j-1}W_f \\
&\quad + \sigma\left(o_j\right)\tanh'\left(c_j\right)\sigma'\left(i_j\right)\tanh\left(z_j\right)W_i + \sigma\left(o_j\right)\tanh'\left(c_j\right)\sigma\left(i_j\right)\tanh'\left(z_j\right)W_z \\
\frac{\partial h_{j-1}}{\partial x_{j-1}} &= \sigma'\left(o_{j-1}\right)\tanh\left(c_{j-1}\right)U_o + \sigma\left(o_{j-1}\right)\tanh'\left(c_{j-1}\right)\sigma'\left(f_{j-1}\right)c_{j-2}U_f \\
&\quad + \sigma\left(o_{j-1}\right)\tanh'\left(c_{j-1}\right)\sigma'\left(i_{j-1}\right)\tanh\left(z_{j-1}\right)U_i + \sigma\left(o_{j-1}\right)\tanh'\left(c_{j-1}\right)\sigma\left(i_{j-1}\right)\tanh'\left(z_{j-1}\right)U_z \\
\frac{\partial h_j}{\partial x_j} &= \sigma'\left(o_j\right)\tanh\left(c_j\right)U_o + \sigma\left(o_j\right)\tanh'\left(c_j\right)\sigma'\left(f_j\right)c_{j-1}U_f \\
&\quad + \sigma\left(o_j\right)\tanh'\left(c_j\right)\sigma'\left(i_j\right)\tanh\left(z_j\right)U_i + \sigma\left(o_j\right)\tanh'\left(c_j\right)\sigma\left(i_j\right)\tanh'\left(z_j\right)U_z
\end{aligned}
$$

where $o_j$ denotes the value in the output gate at $j$-th timestep (similar notations are used for input gate $(i_j)$, forget gate $(f_j)$ and cell value $(z_j)$) and "non-matrix" terms actually represent diagonal matrices corresponding to these terms. Under the assumption that $h_0=0$, we then make use of the results in (Miller & Hardt, 2018) stating that $\|c_t\|_\infty$ is bounded for all $t$. By applying $l_\infty$-norm sub-multiplicativity and triangle inequality, we can show that

$$
\frac{\partial h_j}{\partial h_{j-1}}\frac{\partial h_{j-1}}{\partial x_{j-1}} = M\frac{\partial h_j}{\partial x_j} + N
$$

with

$$
\begin{aligned}
\|M\|_\infty &\le 1/4\,\|W_o\|_\infty + 1/4\,\|W_f\|_\infty\,\|c_j\|_\infty + 1/4\,\|W_i\|_\infty + \|W_z\|_\infty = B_m \\
\|N\|_\infty &\le 1/16\,\|W_o U_i\|_\infty + 1/16\,\|W_o U_f\|_\infty\left(\|c_j\|_\infty + \|c_{j-1}\|_\infty\right) + 1/4\,\|W_o U_z\|_\infty \\
&\quad + 1/16\,\|W_i U_o\|_\infty + 1/16\,\|W_i U_f\|_\infty\left(\|c_j\|_\infty + \|c_{j-1}\|_\infty\right) + 1/4\,\|W_i U_z\|_\infty \\
&\quad + 1/16\left(\|W_f U_o\|_\infty + 1/16\,\|W_f U_i\|_\infty + 1/4\,\|W_f U_z\|_\infty\right)\left(\|c_j\|_\infty + \|c_{j-1}\|_\infty\right) \\
&\quad + 1/4\,\|W_z U_o\|_\infty + 1/4\,\|W_z U_f\|_\infty\left(\|c_j\|_\infty + \|c_{j-1}\|_\infty\right) + 1/4\,\|W_z U_i\|_\infty \\
&= B_n
\end{aligned}
$$

By applying $l_\infty$-norm sub-multiplicativity and triangle inequality,

$$
\begin{aligned}
c_{i-1,t} &= \left\|\prod_{j=i+1}^{t}\frac{\partial h_j}{\partial h_{j-1}}\frac{\partial h_i}{\partial h_{i-1}}\frac{\partial h_{i-1}}{\partial x_{i-1}}\right\|_\infty \\
&= \left\|\prod_{j=i+1}^{t}\frac{\partial h_j}{\partial h_{j-1}}\left(\frac{\partial h_i}{\partial x_i}m + n\right)\right\|_\infty \\
&\le B_m c_{i,t} + B_n\prod_{j=i+1}^{t}\left\|\frac{\partial h_j}{\partial h_{j-1}}\right\|_\infty
\end{aligned}
$$

As LSTM is $\lambda$-contractive with $\lambda < 1$ in the $l_\infty$-norm (readers are recommended to refer to (Miller & Hardt, 2018) for proof), which implies $\left\|\frac{\partial h_j}{\partial h_{j-1}}\right\|_\infty < 1$, $B_n \prod_{j=i+1}^{t} \left\|\frac{\partial h_j}{\partial h_{j-1}}\right\|_\infty \to 0$ as $t - i \to \infty$. For $t - i < \infty$, under the assumption that $\frac{\partial h_j}{\partial x_j} \neq 0$, we can always find some value $B < \infty$ such that $c_{i-1,t} \leq B c_{i,t}$. For $t - i \to \infty$, $\lambda_c \to B_m$. That is, $\lambda_c = \max(B_m, B)$.

## D  Proof of theorem 1

*Proof.* Given that $\lambda_c c_{i,t} \geq c_{i-1,t}$ with some $\lambda_c \in \mathbb{R}^+$, we can use $c_{t,t} \lambda_c^{t-i}$ as the upper bound on $c_{i,t}$ with $i = \overline{1, t}$, respectively. Therefore,

$$f(0) \leq \sum_{t=1}^{T} c_{t,T} \leq c_{T,T} \sum_{t=1}^{T} \lambda_c^{T-t} = f(\lambda_c)$$

where $f(\lambda) = c_{T,T} \sum_{t=1}^{T} \lambda^{T-t}$ is continuous on $\mathbb{R}^+$. According to intermediate value theorem, there exists $\lambda \in (0, \lambda_c]$ such that $c_{T,T} \sum_{t=1}^{T} \lambda^{T-t} = \sum_{t=1}^{T} c_{t,T}$. $\qquad \square$

## E  Proof of theorem 2

*Proof.* According to Theorem 1, there exists some $\lambda_i \in \mathbb{R}^+$ such that the summation of contribution stored between $K_i$ and $K_{i+1}$ can be quantified as $c_{K_{i+1}, K_{i+1}} \sum_{t=K_i}^{K_{i+1}} \lambda_i^{K_{i+1}-t}$ (after ignoring contributions before $K_i$-th timestep for simplicity). Let denote $P(\lambda) = \sum_{t=K_i}^{K_{i+1}} \lambda^{K_{i+1}-t}$, we have $P'(\lambda) > 0, \forall \lambda \in \mathbb{R}^+$. Therefore, $P(\lambda_i) \geq P\left(\min_i(\lambda_i)\right)$. Let $C = \min_i(c_{i,i})$ and $\lambda = \min_i(\lambda_i)$, the average contribution stored in a MANN has a lower bound quantified as $I_\lambda$, where $I_\lambda = C \dfrac{\sum_{t=1}^{K_1} \lambda^{K_1-t} + \sum_{t=K_1+1}^{K_2} \lambda^{K_2-t} + ... + \sum_{t=K_{D-1}+1}^{K_D} \lambda^{K_D-t} + \sum_{t=K_D+1}^{T} \lambda^{T-t}}{T}$. $\qquad \square$

## F  Proof of theorem 3

*Proof.* The second-order derivative of $f_\lambda(x)$ reads:

$$f_\lambda''(x) = -\frac{(\ln \lambda)^2}{1 - \lambda} \lambda^x \tag{11}$$

We have $f_{\lambda}''(x) \leq 0$ with $\forall x \in \mathbb{R}^+$ and $1 > \lambda > 0$, so $f_\lambda(x)$ is a concave function. Thus, we can apply Jensen inequality as follows:

$$\frac{1}{D+1} \sum_{i=1}^{D+1} f_\lambda(l_i) \leq f_\lambda\left(\sum_{i=1}^{D+1} \frac{1}{D+1} l_i\right) = f_\lambda\left(\frac{T}{D+1}\right) \tag{12}$$

Equality holds if and only if $l_1 = l_2 = ... = l_{D+1} = \frac{T}{D+1}$. We refer to this as *Uniform Writing* strategy. By plugging the optimal values of $l_i$, we can derive the maximized average contribution as follows:

$$I_\lambda max \equiv g_\lambda(T, D) = \frac{C(D+1)}{T} \left(\frac{1 - \lambda^{\frac{T}{D+1}}}{1 - \lambda}\right) \tag{13}$$

When $\lambda = 1$, $I_\lambda = \frac{C}{T} \sum_{i=1}^{D+1} l_i = C$. This is true for all writing strategies. Thus, Uniform Writing is optimal for $0 < \lambda \leq 1$. $\qquad \square$

We can show that this solution is also optimal for the case $\lambda > 1$. As $f_\lambda''(x) > 0$ with $\forall x \in \mathbb{R}^+; \lambda > 1$, $f_\lambda(x)$ is a convex function and Eq. (12) flips the inequality sign. Thus, $I_\lambda$ reaches its minimum with Uniform Writing. For $\lambda > 1$, minimizing $I_\lambda$ is desirable to prevent the system from diverging.

We can derive some properties of function $g$. Let $x = \frac{D+1}{L}$, $g_\lambda(L, D) = g_\lambda(x) = Cx(\frac{\lambda^{\frac{1}{x}} - 1}{\lambda - 1})$. We have $g_\lambda'(x) = C\lambda \left(1 - \lambda^{\frac{1}{x}}\right)(x - \ln \lambda) > 0$ with $0 < \lambda \leq 1, \forall x \geq 0$, so $g_\lambda(T, D)$ is an increasing function if we fix $T$ and let $D$ vary. That explains why having more memory slots helps improve memorization capacity. If $D = 0$, $g_\lambda(T, 0)$ becomes E.q (5). In this case, MANNs memorization capacity converges to that of recurrent networks.

## G    SUMMARY OF SYNTHETIC DISCRETE TASK FORMAT

| Task | Input | Output |
|---|---|---|
| Double | $x_1 x_2 ... x_T$ | $x_1 x_2 ... x_T x_1 x_2 ... x_T$ |
| Copy | $x_1 x_2 ... x_T$ | $x_1 x_2 ... x_T$ |
| Reverse | $x_1 x_2 ... x_T$ | $x_T x_{T-1} ... x_1$ |
| Add | $x_1 x_2 ... x_T$ | $\frac{x_1 + x_{T-1}}{2} \frac{x_2 + x_{T-2}}{2} ... \frac{x_{\lfloor T/2 \rfloor} + x_{\lceil T/2 \rceil}}{2}$ |
| Max | $x_1 x_2 ... x_T$ | $\max(x_1, x_2) \max(x_3, x_4) ... \max(x_{T-1}, x_T)$ |

Table 5: Synthetic discrete task's input-output formats. $T$ is the sequence length.

## H    UW PERFORMANCE ON BIGGER MEMORY

| Model | $N_h$ | Copy (L=500) |
|---|---|---|
| DNC | 128 | 24.19% |
| DNC+UW | 128 | 81.45% |

Table 6: Test accuracy (%) on synthetic copy task. MANNs have 50 memory slots. Both models are trained with 100,000 mini-batches of size 32.

## I    MEMORY OPERATING BEHAVIORS ON SYNTHETIC TASKS

In this section, we pick three models (DNC, DNC+UW and DNC+CUW) to analyze their memory operating behaviors. Fig. 4 visualizes the values of the write weights and read weights for the copy task during encoding input and decoding output sequence, respectively. In the copy task, as the sequence length is 50 while the memory size is 4, one memory slot should contain the accumulation of multiple timesteps. This principle is reflected in the decoding process in three models, in which one memory slot is read repeatedly across several timesteps. Notably, the number of timesteps consecutively spent for one slot is close to 10-the optimal interval, even for DNC ( Fig. 4(a)), which implies that the ultimate rule would be the uniform rule. As UW and CUW are equipped with uniform writing, their writing patterns follow the rule perfectly. Interestingly, UW chooses the first written location for the final write (corresponding to the <eos> token) while CUW picks the last written location. As indicated in Figs. 4(b) and (c), both of them can learn the corresponding reading pattern for decoding process, which leads to good performances. On the other hand, regular DNC fails to learn a perfect writing strategy. Except for the timesteps at the end of the sequence, the timesteps are distributed to several memory slots while the reading phase attends to one memory slot repeatedly. This explains why regular DNC cannot compete with the other two proposed methods in this task.

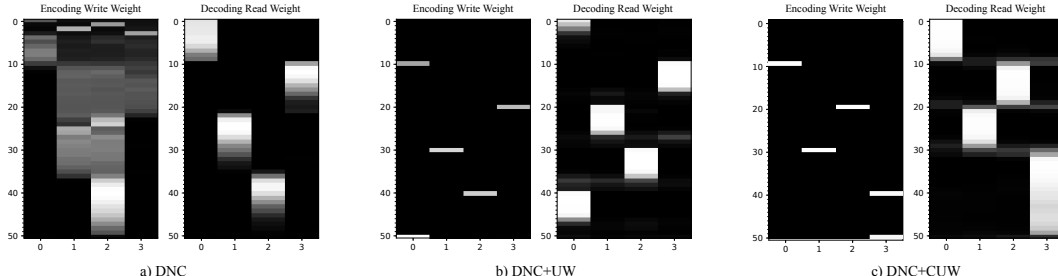

Figure 4: Memory operations on copy task in DNC (a), DNC+UW (b) and DNC+CUW(c). Each row is a timestep and each column is a memory slot.

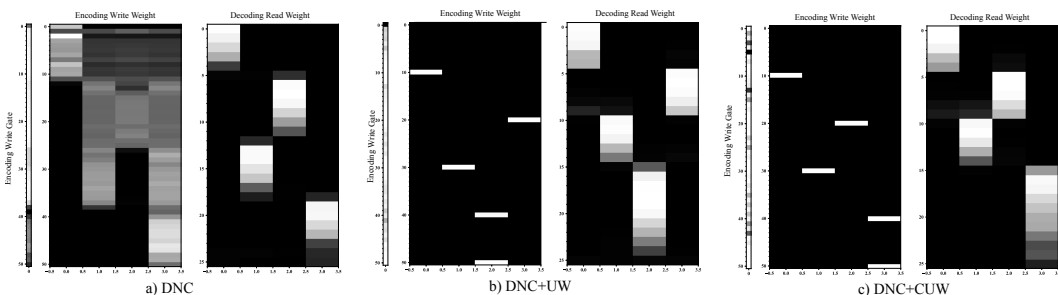

Figure 5: Memory operations on max task in DNC (a), DNC+UW (b) and DNC+CUW(c). Each row is a timestep and each column is a memory slot.

For the max task, Fig. 5 displays similar visualization with an addition of write gate during encoding phase. The write gate indicates how much the model should write the input at some timestep to the memory. A zero write gate means there is no writing. For this task, a good model should discriminate between timesteps and prefer writing the greater ones. As clearly seen in Fig. 5(a), DNC suffers the same problem as in copy task, unable to synchronize encoding writing with decoding reading. Also, DNC's write gate pattern does not show reasonable discrimination. For UW (Fig. 5(b)), it tends to write every timestep and relies on uniform writing principle to achieve write/read accordance and thus better results than DNC. Amongst all, CUW is able to ignore irrelevant timesteps and follows uniform writing at the same time (see Fig. 5(c)).

## J    Visualizations of model performance on sinusoidal regression tasks

We pick randomly 3 input sequences and plot the output sequences produced by DNC, UW and CUW in Figs. 6 (clean) and 7 (noisy). In each plot, the first and last 100 timesteps correspond to the given input and generated output, respectively. The ground truth sequence is plotted in red while the predicted in blue. We also visualize the values of MANN write gates through time in the bottom of each plots. In irregular writing encoding phase, the write gate is computed even when there is no write as it reflects how much weight the controller puts on the timesteps. In decoding, we let MANNs write to memory at every timestep to allow instant update of memory during inference.

Under clean condition, all models seem to attend more to late timesteps during encoding, which makes sense as focusing on late periods of sine wave is enough for later reconstruction. However, this pattern is not clear in DNC and UW as in CUW. During decoding, the write gates tend to oscillate in the shape of sine wave, which is also a good strategy as this directly reflects the amplitude of generation target. In this case, both UW and CUW demonstrate this behavior clearer than DNC.

Under noisy condition, DNC and CUW try to follow sine-shape writing strategy. However, only CUW can learn the pattern and assign write values in accordance with the signal period,

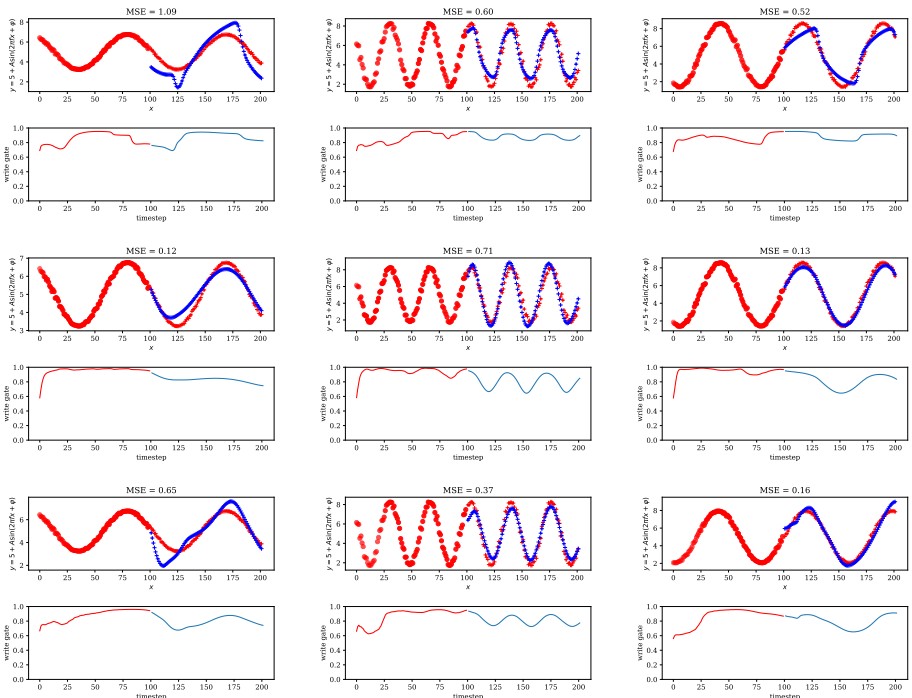

Figure 6: Sinusoidal generation with clean input sequence for DNC, UW and CUW in top-down order.

which helps CUW decoding achieve highest accuracy. On the other hand, UW choose to assign write value equally and relies only on its maximization of timestep contribution. Although it achieves better results than DNC, it underperforms CUW.

## K  Comparison with non-recurrent methods in flatten image classification task

| Model | MNIST | pMNIST |
|---|---|---|
| DNC+CUW | 99.1 | 96.3 |
| The Transformer[⋆] | 98.9 | 97.9 |
| Dilated CNN[♦] | 98.3 | 96.7 |

Table 7: Test accuracy (%) on MNIST, pMNIST. Previously reported results are from (Vaswani et al., 2017)[⋆] and (Chang et al., 2017)[♦].

## L  Details on document classification datasets

| Dataset | Classes | Average lengths | Max lengths | Train samples | Test samples |
|---|---|---|---|---|---|
| IMDb | 2 | 282 | 2,783 | 25,000 | 25,000 |
| Yelp Review Polarity (Yelp P.) | 2 | 156 | 1,381 | 560,000 | 38,000 |
| Yelp Review Full (Yelp F.) | 5 | 158 | 1,381 | 650,000 | 50,000 |
| AG's News (AG) | 4 | 44 | 221 | 120,000 | 7,600 |
| DBPedia (DBP) | 14 | 55 | 1,602 | 560,000 | 70,000 |
| Yahoo! Answers (Yah. A.) | 10 | 112 | 4,392 | 1,400,000 | 60,000 |

Table 8: Statistics on several big document classification datasets

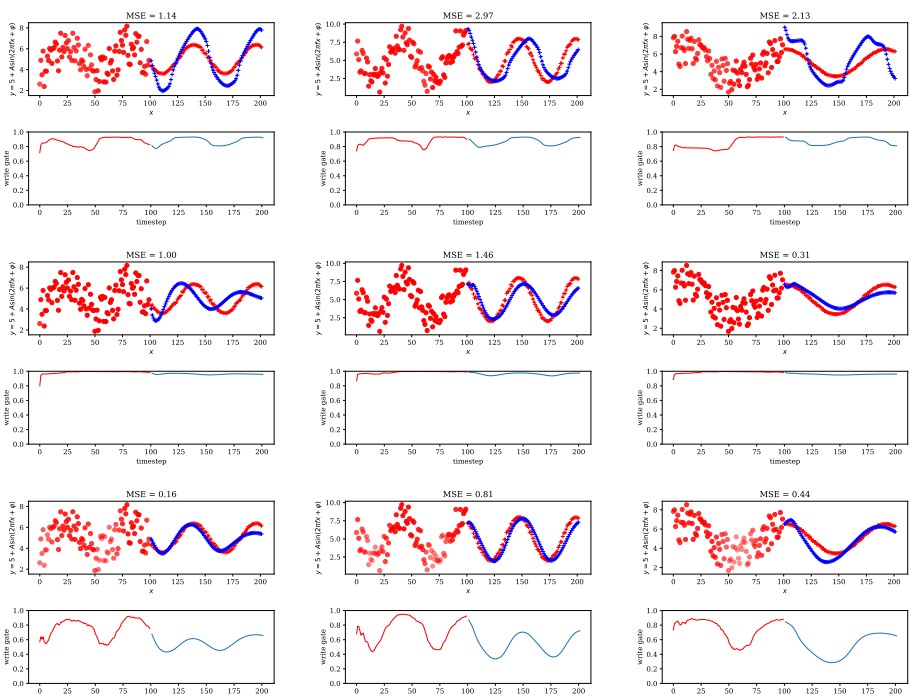

Figure 7: Sinusoidal generation with noisy input sequence for DNC, UW and CUW in top-down order.

## M  DOCUMENT CLASSIFICATION DETAILED RECORDS

| Model | | AG | IMDb | Yelp P. | Yelp F. |
|---|---|---|---|---|---|
| UW | 1 | 93.42 | **91.39** | **96.39** | 64.89 |
| | 2 | 93.52 | 91.30 | 96.31 | 64.97 |
| | 3 | **93.69** | 91.25 | 96.39 | **65.26** |
| | Mean/Std | 93.54±0.08 | 91.32±0.04 | 96.36±0.03 | 65.04±0.11 |
| CUW | 1 | 93.61 | 91.26 | **96.42** | **65.63** |
| | 2 | **93.87** | 91.18 | 96.29 | 65.05 |
| | 3 | 93.70 | **91.32** | 96.36 | 64.80 |
| | Mean/Std | 93.73±0.08 | 91.25±0.04 | 96.36±0.04 | 65.16±0.24 |

Table 9: Document classification accuracy (%) on several datasets reported for 3 different runs. Bold denotes the best records.

