# OpenReview forum: "Learning to Remember More with Less Memorization"
_ICLR.cc/2019/Conference_

### Official Review · AnonReviewer2 · 2018-11-01
**[Updated] Original title: ACT is a crucial missing baseline.**

**Rating:** 7
**Confidence:** 4

**Review:**

This paper deals with Memory Augmnted Neural Networks (MANN) and introduces an algorithm which allows full writes to the dense memory to be only exectued every L timesteps. The controller produces a hidden output at most timestps, whih is appended to a cache. Every L steps, soft attention is used to combine this cache of N hidden states to a single one, and then this is used as the input hidden state for the controller, with the outputs performing a write in the full memory M, along with clearing the cache.

The authors first derive "Uniform Writing" (UW) which updates the memory at regular intervals instead of every timestep. The derivation is based on the "contribution" which is norm of the gradient of some input timestep to some hidden state (potentially at a different timestep). I am not clear on whether this terminology for the quantity is novel, if this is the case maybe the authors should state this more clearly. UW says that if all timesteps are equally important, and only D writes can be made in a sequence of length T, then writes should be done every T/(D+1) steps. I have not checked the proof in detail but this seems reasonable that it would maximise the contribution quantity introduced. I am less clear on whether this is obviously the right thing to do - sometimes this value is referred to in relation to information, but that term does not strictly seem to be being used in the information theory sense (no mention of bits or nats anywhere). Regardless, as the authors point out, in real problems there are obviously timesteps which have less or no useful information, and clearly UW is mostly defined in order to build towards CUW.

CUW expands on UW by adding the cache of different hidden states, and using soft attention over them. This feels like a reasonable step, although I would presume there are times when the L hidden states were collected over timesteps with no information, and so the resulting write is not that useful, and times when all of hte L timesteps contain different useful information. In these circumstances it seems like the problem of getting the *useful* information into the memory is still present, as the single write done with the averaged hidden state will need to contain lots of information, which may be more ideal written with several timesteps.

The experiments are well described and overall the paper seems reproducable. The standard toy datasets of copy / reverse / sinusoid are used. The results are interesting - regular DNC with memory size 50 performs surprisingly badly on clean Sinusoid, my guess would be that with hyperparameter tuning this could be improved upon. I'm not sure that using exactly the same hyperparameters for a wide variety of models is appropriate - even with optimizers like Adam and RMSProp, I would want to see at least some sweeping for the best hyperparams, and then graphs like figure 3 should show error bars averaged across multiple runs with the best per-model hyperparameters. However, The DNC with CUW seems to perform well across all synthetic tasks.

There is no mention of Adaptive Computation Time/ACT (Graves, https://arxiv.org/abs/1603.08983) throughout the paper, which is surprising considering Alex Graves' models form two of the baselines used throughout the paper. ACT aims to execute an RNN a variable number of times, usually to do >1 timestep of processing for a single timestep of input. In the context of this paper, I believe it could be adapted to do either zero or one steps of computation per timestep, and that would yield a very comparable network where the LSTM controller always executes, and writes to the memory only happen sometimes. Given that it allows a learned process to decide whether to write, as opposed to having a fixed L which separates full writes, this should have the potential to outperform CUW, as it could learn that at certain times, writes must happen at every step. In my view ACT is attempting to solve essentially the same problem as this paper, so it should either be included as a baseline, or the manuscript should be updated to explain why this is not an appropriate comparison.


I think this is an interesting paper, trying to make progress on an important problem. The results look good, but I can only give a borderline score due to missing ACT numbers, and a few other unclear points. The addition of ACT experiments, and error bars on certain results, would change my mind here.


Notes:

"No solution has been proposed to help MANNs handle ultra long sequence" - (Rae et al 2016) is an attempt to do this, by improving the complexity of reads / writes. This allows bigger memory and longer sequences to be processed.

"Current MANNS only support dense writing" - presumably this means dense as in 'every timestep', but this terminology is overloaded - you could consider NTM / DNC as doing dense writing, and then work of Rae et al 2016 doing sparse writing.

In my experience training these kind of RNNs can have reasonably high variance across seeds - figures 2 & 3 should have error bars, and especially Table 4 as that contains the most important results. Getting 99 percent accuracy when previous SOTA is only 0.1% lower is only really meaningful if the standard deviation across seeds is very small.

Appendix A: the 'by induction' result - I believe there is an error, it should be:

h_t = \sigma_{i=1}^t U_{t-i}W x_i + C

As W is applied to inputs, before the repeated applications of U? I believe the rest of the derivation still holds the same, after the correction.

---

> ### Author Response · Authors · 2018-11-13
> **Response to reviewer 2**
>
> Thank you for your helpful comments. We would like to address your concerns as follows,
>
> 1. To the best of our knowledge, this is the first time the norm \left\Vert \frac{\partial h_{t}}{\partial x_{i}}\right\Vert  is used in measuring memorization capacity of a recurrent network, which can be regarded as a novelty. We have made this point clearer in this revision.
>
> 2. Regarding to your concern on the validity of our quantity, we agree that there is no direct link to the “information” in information theory sense. Actually, we approached the problem from a different viewpoint. In recurrent networks, one often makes prediction based on h_{t}, which can be considered as a function of timestep inputs, i.e, h_{t}=f\left(x_{1},x_{2},...,x_{t}\right). One way to measure how much an input x_{i} contributes to h_{t} is to calculate \left\Vert \frac{\partial h_{t}}{\partial x_{i}}\right\Vert . If the norm equals zero, h_{t} is constant w.r.t x_{i}. That is, h_{t} does not contain any “information” on x_{i}. A bigger norm implies more influence of x_{i} on h_{t}. As we cannot know in advance which (or all) inputs are required for h_{t} to make good predictions, a reasonable policy is to ensure, on average, all of these norms do not approach zeros, which leads to a maximization problem as shown in our paper. Our empirical results have demonstrated the benefit of following this principle, which enhances our belief that this is the right thing to do.
>
> 3. We have added hyper-parameter tuning for the Sinusoidal Regression task and updated the results in this revision.
>
> 4. Your reasoning on CUW operation is correct. However, even when writing every timestep can capture several important events, this behavior will finally lead to overwriting and loss of information because of finite memory size. Therefore, we believe a balance between following a generic principle and allowing a flexible learning mechanism is beneficial. CUW is one possible solution and we need further investigation to find better writing strategies in future work.
>
> 5. We are aware of ACT [1] and decided not to include it in our references as the goal of our paper and ACT are totally different. While our paper aims to answer the question “when to write to the memory”, ACT aims to answer the question “how many computational steps to take”. However, we agree that if adapted as the reviewer suggests, ACT supports a simple mechanism of learning to write or not to write and should be cited in related works (updated in this revision).
> Nevertheless, the adapted ACT is somehow equivalent to LSTM controller with DNC memory module. When the number of computation steps n is either 0 or 1, ACT mean-field approximation is equivalent to multiplying the state with a learnable gate and we think the output gate in LSTM already supports that. Extending to memory level, this is equivalent to multiplying the writing weight with a learnable gate (if the gate equals zero, there is no writing at that timestep). DNC is equipped with a write gate g_{t}^{w}, which executes the same function (see Eq. (2) in [2]). Hence, we strongly believe that an ACT baseline is unnecessary as DNC is capable of deciding whether to write at each timestep. In theory, DNC itself can learn uniform writing strategy. However, in practice, it is very hard to learn a particular writing scheme without any guidance. This emphasizes the importance of searching for a writing policy that is guided by optimal principles instead of trying to learn everything end-to-end. The fact that DNC+CUW outperforms DNC in various experiments further validates our argument.
>
> 6. Our original claim is “no solution has been proposed to help MANNs handle ultra-long sequences given limited memory”. The authors in [3] aim to learn longer sequences by scaling the memory size, which is not conditioned on our limited memory setting. To make our claim less confusing, we have added another sentence to differentiate between our work and [3].
>
> 7. We admit the term “dense writing” is confusing, and thank you for pointing it out. The same confusion may apply to the term “sparse writing”. Therefore, we have replaced the two terms with “regular writing” and “irregular writing”, respectively.
>
> 8. We agree with you on the addition of error bars on Figs. 2, 3 and Table 4. We have collected and included these statistics for the synthetic tasks in this version of our paper. We have been working on the document classification task and hope that we can include error bars for this task before the revision deadline.
>
> 9. Your comment on the order of U and W is correct. We have fixed that in this revision. Thank you for your detailed reading.
>
> [1] Graves et al., Adaptive computation time for recurrent neural networks. arXiv preprint arXiv:1603.08983 (2016)
> [2] Graves et al., Hybrid computing using a neural network with dynamic external memory. Nature, 2016.
> [3] Rae et al., Scaling memory-augmented neural networks with sparse reads and writes. NIPS'16

---

### Official Review · AnonReviewer1 · 2018-11-03
**High quality piece of research**

**Rating:** 8
**Confidence:** 4

**Review:**

This paper investigates the average contribution of a sequence input to the contents of memory and derives a simple scheme to maximize the information content in memory, which is essentially to write at uniformly spaced intervals. Furthermore they present an attention-based version, where the network caches all hidden states in an interval and selects the hidden state to store via attention.

The paper is very well written and has a nice balance of relevant theoretic motivation and experiments. Furthermore the question that the authors are tackling --- how should we compress information into external memories --- feels important and under-explored. The fact that the resulting scheme is simple is nice, because it's easy for people to try, and it now has some motivation beyond a heuristic decision.

I think this paper will have impact in opening up more comprehensive research into the reduction of redundancy in the external memories of neural networks, and also could be instantly impactful for people using DNCs and NTMs --- especially since we see the incorporation of UW / CUW can help bridge the gap (or even surpass) LSTMs for the modeling of natural data. As such I think it is a clear accept.

---

Comments to the authors:

The results in Figure 2 (c) I think are misleading. The NTM with an RNN controller can solve this task, the limit of 10,000 steps implies that the model may converge to some 50% value with 14 slots but I am absolutely certain that the NTM + RNN controller would converge in 10,000 steps with a careful tuning of gradient clipping and learning rate. I think this is basically a false result. Furthermore I would like to really know what the best final performance of the models are on this task once converged, it's not clear if 10,000 steps was enough.

For equation (9), was it necessary to construct the attention weights in this way? How much better was it to a direct softmax query: softmax(h_{t-1}^T d_j)? If you are backpropagating through the attention then the network can shape the hidden states to facilitate the relevant attention, as well as contain the information.

In the second paragraph of S2.2.2 you have "a_{t, j} is the attention score" but you should have "\alpha_{t, j} is the attention score".

Table 3: just include the Transformer results in the table!? The reasoning to exclude it is not really coherent.

It would have been nice (and would raise my score) to see the UW scheme operating with a large(ish) number of memory slots.

---

> ### Author Response · Authors · 2018-11-13
> **Response to reviewer 1**
>
> Reviewer 1:
>
> Thank you for your constructive comments. We would like to address your concerns as follows,
>
> 1. We are aware of the unexpected performance of NTM+RNN with 14 memory slots. It should be noted that the results reported in Table 2 (c) are the averages of accuracy over multiple running times, in which NTM+RNN converges sometimes but not always under our training setting. To demonstrate that our UW is helpful under various training settings, we have reassessed the models with different learning rates (0.001,0.0001) and gradient clipping (1,5,10). We have reported the mean performance with error bars in the updated manuscript.
>
> 2. There are two reasons for stopping after 10,000 training steps. First, the learning curves look stable and show no promise to gain big improvement around 10,000 steps. Second and more importantly, in our synthetic tasks, training with more steps means the models access to more training data and are likely to gradually overfit. This behavior is clearer when the number of memory slots increases where both regular and uniform writing often solve the synthetic tasks perfectly if they are trained with unlimited data. We want to avoid that setting and focus on measuring the performance on unseen test data given a moderate amount of training samples as in reality the training data is very limited.
>
> 3. Eq. (9) is inspired by Bahdanau attention [1] (the “concat”) in which the alignment model is implemented as a neural network with additional parameters. We think this mechanism will be more flexible than your direct softmax query (the “dot”) as the attention does not restrict to similarity. Also, we want to utilize read values from the memory, which may give useful information for the attention. The “concat” form naturally suits our purpose.
>
> 4. Thank for pointing out the typo in S.2.2.2. We have fixed the typo in this revision.
>
> 5. In Table 3, we aim to validate our method against other recurrent baselines in their capacity to memorize efficiently. The Transformer, on the other hand, accesses to all timesteps and thus, does not need manage memorization. For completeness, we have now included the results of the Transformer, together with the Dialated CNN, as non-recurrent baselines in Appendix I in this revision.
>
> 6. We have conducted the copy task with bigger memory (number of memory slots=50 and sequence length=500). At this moment, after 40,000 batches, DNC +UW's best validation accuracy is 38.1% while DNC's is 17.2%. The final results will be put in Appendix in the next revision.
>
> [1] Bahdanau et al., Neural Machine Translation by Jointly Learning to Align and Translate. ICLR'15

---

> > ### Comment · AnonReviewer1 · 2018-11-25
> > **Response to revision**
> >
> > Thanks for your responses and paper revisions. I still agree this is a nicely conducted piece of research and will retain my score of 8.
> >
> > Re. (1) & (2) I see, in most cases when people perform the copy task they train on programatically generated sequences that essentially cannot be overfit to (e.g. because the domain of possible sequences to copy is very very large). In this setting more than 10,000 steps is usually useful, and I would expect the NTM to always converge to low loss-error eventually. However I now understand your setup a little better, thanks for clarifying.
> >
> > Re (3). Yes but my point is that it was jarring not to see the Bahdanau attention, which uses dot-product as a distance metric, or at least have an ablation where you show that your more general MLP attention performs significantly better. But I don't feel strongly.
> >
> > Re (5) Well I'm sure you would have included it in the main table if your model had better performance ;-)

---

### Official Review · AnonReviewer3 · 2018-11-15
**Very interesting and well written paper on augmented memories**

**Rating:** 7
**Confidence:** 3

**Review:**

This paper looks at ways to improve memory-writing in memory augmented neural networks. Authors proposed two methods to compare against "regular writing" method as well as compare against each other, namely "uniform writing" and "cached uniform writing". Latter one attempts to utilize a small size memory efficiently by introducing memory overwriting in other words "forgetting".

Authors started with a very interesting section (namely section 2.1.1) and presented a theoretical formulation of "remembering" capability of RNNs, which is fundamental to this work and I really liked it that they did not jump to the proposed methods right away and instead focused on something very fundamental. Authors presented details of the proposed methods very well, and evaluated them on simple tasks such as "double task", "synthetic reasoning", etc. as well as on more challenging/real tasks such as "document classification" or "image recognition task from MNIST". I really liked the fact that the paper looked at different tasks instead of going with one. Results are convincing overall, especially for CUW. One thing that will improve the paper is the analysis part.

Due to having 5+ tasks in the results section, I got the feeling that it is hard to follow the analysis presented by authors within each task as well as across tasks. Also, in some tasks analysis is quite limited. It would be great for authors to zoom into the memory write operations in each task (e.g., taking a diff between RW and URW for example and see how memory changes and more importantly how "remember" capability changes) and provide more stats on these, and do this across tasks in one section rather than in different sections allocated for each task. Also, analysis in more realistic tasks (e.g., document classification) can be extended as well, rather than only comparing against state-of-the-art methods in terms of final metric.

While reviewing the paper, I couldn't help asking why larger memories were not tried. I can see the motivation of trying to use smaller augmented memory, however experimentation around slightly larger augmented memories will be useful for the audience to draw some conclusions. Especially I'm curious about the effect of memory size on accuracy in tasks like image recognition or document classification.

---

> ### Author Response · Authors · 2018-11-25
> **Response to reviewer 3**
>
> Thank you for your constructive comments. We would like to address your concerns as follows,
>
> 1. In Appendix H of the last revision, we included an analysis on memory writing for the Sinusoidal Regression task, which gives some insight into the difference between DNC, UW and CUW's writing strategy. We have conducted other analyses on some synthetic tasks and put the results in Appendix I in this revision. We hope that together with the one in the Sinusoidal Regression task (Appendix J in this revision), these inspections provide readers with better understanding on operations inside memory under regular and irregular writing policies. For real-world tasks, it is very hard to visualize the memory operations with realistic data as they are are very long sequences with unknown properties. Hence, we will leave that for future works.
>
> 2. We agree with you on the addition of larger memory experiments. We have included one in Appendix H in this revision. For flatten image recognition tasks, we tried with different memory sizes {15, 30, 60}  and reported the best results for DNC. For UW and CUW, as increasing memory size makes these methods approach to DNC's performance, their memory sizes are fixed to 15.

---

### Author Response · Authors · 2018-11-25
**Summary of revision**

Dear Reviewers,

Thank you for your insightful comments and valuable suggestions. We have revised our manuscript according to your feedback. In this revision, we have added results for copy task with larger memory (Appendix H) and analyses on memory operations (Appendix I). For document classification running details, we have included the statistics for 4/6 datasets (Appendix M). We have been working with the other two datasets and will put the complete result in the next revision.

---

### Public Comment · (anonymous) · 2019-02-25
**Would you plan to release the source code?**

Congratulations! Would you plan to release the source code?

---

> ### Author Response · Authors · 2019-02-26
> **Source code link**
>
> Thank you for your interest in our paper. We have released a reference implementation at https://github.com/thaihungle/UW-DNC

---

> > ### Public Comment · (anonymous) · 2019-02-28
> > **Thanks**
> >
> > Thanks

---

### Meta-Review · Area_Chair1 · 2018-12-13
**High quality research on memory augmented neural networks**

**Confidence:** 4
**Recommendation:** Accept (Oral)

**Metareview:**

Well-written paper that motivates through theoretical analysis new memory writing methods in memory augmented neural networks. Extensive experimental analysis support and demonstrate the advantages of the new solutions over other recurrent architectures.
Reviewers suggested extension and clarification of the analysis presented in the paper, for example, for different memory sizes. The paper was revised accordingly. Another important suggestion was considering ACT as a baseline. Authors explained clearly why it wasn't considered as a baseline, and updated the paper to include references and explanations in the paper as well.